# “I Go up to the Edge of the Valley, and I Talk to God”: Using Mixed Methods to Understand the Relationship between Gender-Based Violence and Mental Health among Lebanese and Syrian Refugee Women Engaged in Psychosocial Programming

**DOI:** 10.3390/ijerph18094500

**Published:** 2021-04-23

**Authors:** Rassil Barada, Alina Potts, Angela Bourassa, Manuel Contreras-Urbina, Krystel Nasr

**Affiliations:** 1Abaad Resource Center for Gender Equality, Beirut 21133, Lebanon; krystel.nasr@abaadmena.org; 2Gillings School of Public Health, University of North Carolina at Chapel Hill, Chapel Hill, NC 27599, USA; 3The Global Women’s Institute, George Washington University, Washington, DC 20052, USA; apotts@email.gwu.edu (A.P.); abourassa@email.gwu.edu (A.B.); 4World Bank, Washington, DC 20433, USA; mcontrerasurbina@worldbank.org

**Keywords:** gender-based violence, intimate partner violence, mental health, humanitarian, Lebanon, Syrian refugees, gender, LMICs

## Abstract

Lebanon’s intersecting economic and political crises exacerbate complex public health issues among both host and refugee populations. This mixed-methods study by a Lebanese service provider, in partnership with an international research institute, seeks to better understand how experiences of gender-based violence (GBV) and mental health intersect in the lives of Syrian and Lebanese women, and how to better meet these needs. It employs a randomized cross-sectional survey of 969 Abaad service users and focus groups with community members and service providers. There were significant associations between GBV and ill mental health; notably, respondents reporting transactional sex had 4 times the likelihood of severe distress (aOR 4.2; 95% CI 1.2–14.8; *p* ≤ 0.05). Focus groups emphasized less-visible forms of violence, such as emotional violence, and the importance of environmental factors in one’s ability to cope, noting “it always came back to the economy”. Recommendations include providing a more holistic and coordinated approach between GBV, mental health, livelihood, and basic assistance sectors; and sensitive, accessible, and higher-quality mental health services informed by GBV response actors’ experience putting in place survivor-centered programming and made available to both host and refugee community members.

## 1. Introduction

Lebanon’s history of conflict and ongoing crises have had profound social, political, and economic effects on the country and its residents [1,2]. Protracted displacement and increasing vulnerabilities related to economic insecurity, legal status, and severe living conditions further increase the risk of GBV incidences, such as early marriage and intimate partner violence (IPV) [3,4]. These conditions, exacerbated by the COVID-19 pandemic and 2020 Port Explosion, are further exacerbating GBV risks and limiting access to mental health and other services [5]. Some GBV survivors reported having less time and fewer resources to address GBV concerns, and focusing more on their own and their families’ basic needs instead [5].

Humanitarian emergencies also exacerbate rates of ill mental health among refugees and displaced populations, and are compounded by daily stressors of material, financial, and medical insecurity [6,7,8]. This has been reported for both Syrian refugees and host community members with prior diagnoses of mental disorders or conflict-related violence [9,10,11,12,13]. According to several studies, there are linkages between exposure to GBV and the development of psychological distress or mental health illnesses, including suicidal ideation or suicide attempts (in 20–25% of cases), PTSD, or depression [14,15,16,17]. This raises the question of the mental health status of GBV survivors in Lebanon and associations between exposure to GBV and mental and psychological wellbeing in the Lebanese context.

There are several systems in Lebanon that address mental health, including a National Mental Health Programme that aligns with WHO’s Mental Health Gap Action Programme or “mhGAP”, a National Mental Health Strategy that aligned with WHO’s mental health action plan 2013–2020, a National Action Plan for the Mental Health and Psychosocial Support (MHPSS) Response to COVID-19, a MHPSS Task Force, a referral system for Palestinians who need MHPSS services managed by the UN Relief and Works Agency (UNRWA), MHPSS services provided by non-governmental organizations (NGOs), five mental health hospitals, and eight hospital psychiatric wards [18,19,20].

To help fill this gap, we conducted mixed methods research within a socioecological model to explore the link between women’s environmental factors, GBV, and mental wellbeing in Lebanon, and to identify ways in which GBV prevention and response considerations can be better integrated into mental health programs and vice versa, with the aim of better serving GBV survivors with mental health needs. Such high-quality research is possible when service providers (NGOs, women’s rights organizations, and other civil society) and academics collaborate to center research processes on the needs of the affected populations and those who work with them most closely, and subsequently translate findings into action [21].

## 2. Materials and Methods

This mixed-methods study utilized a randomized cross-sectional survey of the Abaad service population, followed by focus group discussions with community members and service providers. The quantitative survey randomly sampled adult women between the ages of 18 and 65 years who have accessed Abaad’s psychosocial support (PSS) services at least once, excluding those who had ever used GBV case management. Case management service-users were excluded to respect privacy and reduce potential retraumatization among survivors of the most severe GBV. However, exclusion of this group may also result in lower reported GBV among study participants than actually occurs in this service population.

Potential participants were chosen randomly from the databases of PSS beneficiaries in the North (Qobbeh) and Northern Bekaa, using the randomization function in Microsoft Excel. Social workers operating in two of Abaad’s Women and Girls Safe Space (WGSS) structures in the sampling areas contacted selected respondents by telephone to explain the study and enquire about interest in participating. If the respondent consented, they set an agreed interview time. Women who were incapacitated or otherwise ineligible (for example, those who could not be reached due to changed or disconnected phone numbers) were recorded as a non-response (NR).

Target sample size was calculated as 824 to provide statistical power based on a 99% confidence level and 4% margin of error. A total of 2217 women (1542 in the Bekaa and 675 in the North) were randomly selected and 981 successfully contacted. This 44% response rate was expected due to the nature of internal displacement of Syrian refugees to a different region in Lebanon or repatriation to Syria, or changes in phone numbers. Additionally, not all have mobile phones (or want to share their personal phone numbers) making these service-users difficult to reach.

Out of 981 women contacted, 975 were recruited (575 in the Bekaa and 400 in the North), giving an enrollment rate of 99%. The high enrollment rate is likely due in part to an existing level of trust between the organization and study participants. Ultimately, 973 participants completed the survey with two participants withdrawing consent. An additional four records were found to have become corrupted during upload and were dropped during initial data cleaning, reducing the final sample to 969.

The data collection team consisted of nine female enumerators (three in the North, six in the Bekaa) from the areas in which they were conducting surveys. Training was conducted by the Global Women’s Institute (GWI) and Abaad team members on GBV and mental health concepts, research ethics and safety, quantitative and qualitative methods (including trauma-informed interviewing), and staff care and supervision, followed by two weeks of piloting the survey instrument. Ethical review and approval were provided by the Institutional Review Board at the Notre Dame University of Lebanon.

Interviews were conducted in-person at Abaad Women and Girls Safe Spaces (WGSS) to ensure confidentiality and safety of participants and enumerators in a setting familiar to the participants (*n* = 585). In cases where participants were unable to travel to the Abaad Center nearest to them, interviews were held at a mutually agreed and confidential space (*n* = 389). A transport stipend was provided to all participants.

The survey instrument (supplement) was designed to categorize the types of violence experienced, assess symptoms of mental health and wellbeing, and gather information relating to access to services, focusing on any barriers or enablers to accessing GBV services. The tool employed in this study was adapted from scales used in humanitarian contexts and those used by Abaad in their ongoing research about GBV, in consultation with representatives from the MHPSS Task Force in Lebanon, and with researchers who have led previous studies on mental health among refugee populations. Each tool is described briefly in Table 1, along with its validity in the Lebanese context. It should be noted that the HESPER and COPE inventories were not employed as assessment scales in this study, but as groupings of factors known to be common among refugee populations and potentially related to GBV and mental health in this population.

Survey questions and skip logic were programmed using the KoBo Toolbox suite developed for use in humanitarian and crisis environments in English, then translated into Arabic by Abaad staff for field use. Enumerators were provided Android tablets with the KoBo Collect app, designed to protect against survey data loss in areas without over-the-air (OTA) data services available. Data collection began in mid-March and was completed by early May to avoid conducting interviews during the Holy Month of Ramadan when the subject matter could be considered inappropriate.

Quantitative data were analyzed using SPSS version 25. Descriptive statistics were obtained for all data groups and bivariate analysis (Pearson chi-square) was conducted to determine statistically significant associations between outcomes of interest (GBV, mental health, and help seeking) and independent variables (demographic, environmental vulnerabilities, and GBV (in relation to mental health)). Multivariate logistic regression was conducted in three models: bivariate logistic regression to further establish the relationship between independent variables and outcomes significant in initial bivariate analysis; multivariate analysis of significant covariates in related blocks (sociodemographic and environmental vulnerabilities); and multivariate analysis of all covariates that remained statistically significant in the second model. The analysis was conducted separately for intimate partner violence (IPV), non-partner violence (NPV), and mental health outcomes.

Following the quantitative survey, focus group discussions (FGDs) were conducted to further explore findings from the quantitative results (e.g., probing about barriers/enablers to accessing services). A total of 15 FGDs (7 in each governorate and 1 on a national level) were conducted with community leaders, service providers, and women community members (number of participants available in Table 2 below):Three with GBV and MH frontline service providers from civil society.Four with community leaders. These groups were separated by gender.Eight with women community members.

Participants of the FGDs were chosen by sending an email to all members of the sexual and gender-based violence (SGBV) and MHPSS task force inviting them to send one relevant staff member (who either specializes in GBV, MH, or both).

Two FGD guidelines were developed, one targeting service providers and community leaders, and another for community members. These included questions and exercises addressing MHPSS, GBV, and service access-related topics (available as Appendix A).

The research team used the thematic analysis to analyze the qualitative data using Dedoose qualitative software. Analysis focused on participants’ conceptualization and understanding of GBV and mental health, and factors that support, or hinder, GBV survivors with mental health needs to access services.

## 3. Results

Overall, 969 women aged 18–65 were included in the sample, about two-thirds of whom identified as Syrian or Palestinian (a fractional minority (*n* = 9) indicated the Palestinian nationality. These women were included in the Syrian group for statistical analysis due to their small number and shared social and legal status as refugees, the majority of whom (*n* = 7) indicated migrating to Lebanon due to the war in Syria) and about one-third Lebanese. The median age of the sample was 38, with Lebanese women being slightly older than Syrian women (42 vs. 36). The vast majority (91%) had never been partnered and 7 in ten were currently married and living with their husbands at the time of the survey. Of those who have been partnered, 98% have children. Fifteen FGDs were conducted with a total of 114 service providers and community leaders and members, split between the Bekaa and North Lebanon. Appendix A including consent forms, tools, and quantitative result tables are available online at the locations noted at the end of this article.

More than half of the surveyed women reported having less than primary education and significantly more Lebanese women indicated having education beyond secondary than Syrian women (17% and 6%, respectively). Paid work is not common among the study sample: only 15% of women reported working for money and paid work was more common among Lebanese women (19%) than Syrian women (13%) who have limited occupations available to them [29].

Nine out of ten (91%) of women reported having serious problems due to one or more environmental vulnerabilities. Food insecurity was the most commonly reported vulnerability (71%), followed by physical health (62%), being separated from family (56%) and safety and security where they live (50%). FGD participants identified lack of access to financial resources—not included in the HESPER inventory—as the main source of vulnerability among both host and refugee communities.

### 3.1. Gender-Based Violence

#### 3.1.1. Forced and Child Marriage

More than one in three women indicated they were married before the age of 18. Marriage before the age of 18 was significantly more common among Syrian women than Lebanese (44% and 30% respectively) (*p* ≤ 0.05) as was forced child marriage (11% and 5%, respectively) (*p* ≤ 0.05) (Table 3, Table 4, Table 5, Table 6 and Table 7). This distinction was between survey participants who reported choosing marriage before the age of 18 and those who reported someone else choosing marriage for them. FGD participants described refugees and lower-income individuals as increasingly vulnerable to child marriage.

#### 3.1.2. Intimate Partner Violence

More than three-quarters of ever-partnered women (77%) in the study reported ever experiencing one or more acts of IPV and half (52%) reported experiencing this violence in the 12 months preceding the survey. Half (52%) of the women experienced lifetime physical and/or sexual IPV with small differences by nationality (46% Lebanese and 52% Syrian) and about 4 in 10 (39%) in the past 12 months (34% Lebanese and 41% Syrian). Qualitative results also highlight the burden of emotional and psychological IPV on women in the north, where some women need to ask their husband’s permission to go to their “neighbor’s or parents’ house, or even to the bathroom”.

Forced child marriage doubled the odds of ever experiencing physical and/or sexual IPV (aOR 2.2; 95%CI 1.3–3.9; *p* ≤ 0.05) compared to those who were married at age 19 or older when adjusting for all other significant factors. This was mirrored in the qualitative results. Similarly, women who reported working for pay or engaging in income generating activities (IGAs) were nearly twice as likely to experience physical and/or sexual IPV (aOR 1.9; 95%CI 1.1–2.6; *p* ≤ 0.05). FGD participants noted that women who engaged in IGAs also noted being vulnerable to labor and sexual exploitation in the workplace. Multiple participants described experiences of violence as a “moral failure” by their families, societies, and the state.

Environmental vulnerabilities were significantly associated with lifetime and current IPV. Women who had serious problems due to physical health or disability were 40% more likely to have ever experienced physical and/or sexual violence (aOR 1.4; 95%CI 1.0–1.9; *p* ≤ 0.05) and twice as likely to have experienced this violence in the past 12 months (aOR 2.0; 95%CI 1.4–2.8; *p* < 0.001). Separation from family and lack of easy and safe access to a clean toilet increased the odds of ever experiencing physical and/or sexual IPV 40–50% (separation aOR 1.4; 95%CI 1.1–1.9; *p* ≤ 0.05; toilet aOR 1.5; 95%CI 1.1–2.0; *p* ≤ 0.05) and lack of safety and security in the community similarly affected the likelihood of IPV in the past 12 months (aOR 1.4, 95%CI 1.0–1.9; *p* < 0.10). Additionally, while FGD participants echoed the importance of safety and security, they most notably linked violence to financial instability; “it all goes back to the economy...the economy creates this violence.

#### 3.1.3. Non-Partner Violence

Non-partner violence of all types occurred at similar rates among Lebanese and Syrian women, suggesting refugee status is not a significant risk factor for NPV in the sample population. Unwanted sexual touching was reported by about 1 in 5 women (19%) and being propositioned for sex in exchange for money or other favors was reported by about 1 in 10 (9%). Fewer than 5% of women reported being forcibly stripped, raped, or having experienced attempted rape. It is worth repeating that our sample excludes women who have engaged Abaad’s case management services (for reasons outlined in the methods section), so these rates of both intimate and non-partner sexual violence are likely lower than they would be among the general population. FGD participants in both the north and Bekaa expressed fear and hesitancy to use public transportation or walk alone after dark, and described harrowing experiences of workplace sexual exploitation, such as employers “offering” sexual favors in lieu of wages.

### 3.2. Mental Health

Psychological distress was measured using the K6+ assessment of psychological distress (Cronbach’s alpha 0.82) using a scale of increasing positivity to describe the frequency of six specific symptoms experienced in the past 30 days, ranging from none of the days (1) to all days (5). A cut score of 19 out of 30 was used as the demarcation between severe distress and mild to moderate distress as a dichotomous result: 19 or higher indicates severe distress, 18 or lower indicates no severe distress. FGD respondents noted symptoms such as “mental fatigue”, “daghet” (an Arabic term for pressure, or a sense of being chronically overwhelmed), social isolation, fear, hypo- and hypersomnia, chronic stress or discomfort, self-doubt, and suicidal ideation, and mental health disorders such as depression and anxiety. While “daghet” is directly translated into “pressure,” it is used colloquially to describe a high burden on one’s mental health while still appearing unphased and strong, as is socially acceptable

#### 3.2.1. Gender-Based Violence

The experience of violence violates women’s sense of safety and security, both in their homes and in their communities. Every dimension of GBV measured by the survey was associated with elevated rates of severe psychological distress. Qualitative results gave insight into the links between different types of GBV, how they may coexist in a relationship, and the ways in which they cause distress. In bivariate analysis, a similar pattern of elevated rates of severe distress emerged for all types of IPV, both lifetime and current: women who experienced emotional or physical and/or sexual IPV in the past 12 months reported significantly higher rates of severe distress (80%, both) than those who have never experienced such violence (66% and 67% respectively) (*p* ≤ 0.05). The mental health effects are also lasting, as women who have ever experienced emotional or physical and/or sexual IPV demonstrated similarly elevated rates of severe distress (80% and 79% respectively) than those who have never experienced such violence (65%, both types) (*p* ≤ 0.05). Participants in all FGD groups illustrated the cyclical relationship between GBV and mental health. Namely, that GBV can cause mentally ill-health symptoms, and that mentally ill-health can exacerbate or cause violence. IPV specifically was reported to be bidirectionally associated with anger, pressure, and irritability. A service provider in the north labelled psychological distress as a “cause and consequence of violence”, echoed by community members in the Bekaa who outlined the inextricable link between GBV and mental health. Survey results highlighted that sexual violence, in particular, has a negative impact on mental health. Among women who have ever experienced sexual assault by a partner or non-partner, 82% demonstrated severe psychological distress compared to 67% of those who never experienced sexual violence (*p* ≤ 0.05). Being propositioned for sex in exchange for money or other favors was a primary factor in both bivariate and multivariate analysis. Women who have ever experienced such a proposition demonstrated the highest rate of severe distress (93% vs. 68% among those who have not) of all women who have experienced any dimension of GBV captured by the survey. Controlling for all other variables, having ever experienced a transactional proposition was associated with 4 times the likelihood of severe distress (aOR 4.1; 95%CI 1.2–14.8; *p* ≤ 0.05).

Forced child marriage was also significantly associated with psychological distress among all women; 89% of women married without consent before age 18 met the criteria for severe distress versus 72% who married later and with consent (*p* ≤ 0.05). Forced child marriage was associated with three times the likelihood of severe distress when controlling for all other family and relationship variables (aOR 3.0; 95%CI 1.9–20.8; *p* ≤ 0.05) but lost significance in the final model. FGD respondents attributed this to the added responsibility of marriage at a young age, conflict between spouses, the burden of bearing children while still a child, and the increased likelihood of experiencing IPV if forced to marry early. A male community leader in the north described the practice as archaic, living on “from Phoenician times”, echoed by a service provider who linked the practice with incidence of depression among women married without consent before the age of 18.

#### 3.2.2. Pressure and Worry in Uncertain Circumstances

Among the study sample, 70% of women demonstrated severe psychological distress and Syrian women showed more severe distress than Lebanese women (80% and 55%, respectively) (*p* ≤ 0.05). This difference was contextualized in FGDs as symptoms stemming from pressure or worry (expressed in Arabic as ضغط or “daghet”) resulting from compromised financial and social status, family stress, uncertainty about the present and future, family separation, and stigma associated with refugee status. It was also partially attributed to some refugees’ living quarters in ITSs; “those who live in a house have completely different mental health, [and] even if you changed the tent and renovate it, it’s still a tent”.

The lived experience of uncertain circumstances was captured in the survey through five dimensions of environmental vulnerabilities included in the HESPER inventory. Having a serious problem due to each defined vulnerability was individually associated with a significantly higher rate of severe psychological distress compared to those who said they did not have a serious problem in that area, ranging from 80% of women who indicated a serious problem due to food insecurity to 85% of women who said they have a serious problem due to safety and security where they live (vs. 51% and 56%, respectively) (*p* ≤ 0.05).

In addition to individual environmental vulnerabilities, the number of vulnerabilities causing a serious problem was significantly associated with increasing rates of severe psychological distress, ranging from 22% among women who had no serious problems due to environmental vulnerabilities to 94% among those who indicated serious problems stemming from all five (*p* ≤ 0.05). Controlling for all other factors, multivariate analysis demonstrated a large and significant increase in odds of severe distress with each additional environmental vulnerability compared to none: one (aOR 5.5; 95%CI 2.1–14.8), two (aOR 8.2; 95%CI 3.3–20.5), three (aOR 14.6; 95%CI 5.9–36.3), four (aOR 25.7; 95%CI 9.0–73.6), and serious problems due to all five (aOR 102.5; 95%CI 12.0–883.3) (*p* < 0.001). In the words of a community member in the north, “financial relief leads to psychological relief”.

### 3.3. Help Seeking

Women were asked about what kind of services they accessed in response to violence and to meet daily needs and factors that supported or created barriers to service uptake. Among women who had experienced GBV, about 7 in 10 said that they accessed services in response to the violence they had experienced. There was no significant difference in violence related service uptake by nationality or region, though a significantly higher proportion of women who accessed violence related services did so in response to physical violence (75% vs. 65% of those who did not experience physical violence) (*p* ≤ 0.05).

Types of services women sought in response to violence varied significantly by nationality and by region. While both Syrian and Lebanese women primarily sought psychosocial support, mental health, or case management, Syrian women had a higher rate of this type of support (78% vs. 69%) (*p* < 0.001). Larger proportions of Lebanese women sought education and vocational training (16%) or other services (12%) compared to Syrian women (8% and 5%, respectively) whereas Syrian women more often sought healthcare services (8% vs. 3%) among Lebanese women. FGD participants reported the need for other formal and informal services, including community and peer-to-peer support, financial assistance from friends and family, escorting affected individuals to needed services, medical and specialized psychiatric/psychological services, livelihoods and shelter, and free education, among others.

The primary barrier to service seeking reported by surveyed women who did not seek services in response to violence was that they simply did not know that services were available (31% overall). This varied between regions with 35% of women in the Bekaa region naming it as their primary barrier vs. 25% in the north (*p* ≤ 0.05). The second most common barrier reported was that women did not believe they needed services to address violence. This was more common in the north (30%) and among Lebanese women (32%) than in the Bekaa region or among Syrian women (24% each) (*p* ≤ 0.05). Though they were both reported by a minority of women as a primary barrier, 5% of women said their husbands would not allow them to seek help and 6% did not seek help out of fear of retaliation. Both of these barriers were reported more often among Lebanese women (7%, not allowed; 8% fear) and in the north (12% not allowed; 8% fear) than among Syrian women (4% either reason) and in the Bekaa region (1% not allowed; 4% fear) (*p* ≤ 0.05).

Qualitative data provided insight into additional barriers to service accessibility, including barriers related to the structure of services, legal, social, awareness, financial, and logistical barriers. Some participants noted that service structure such as a lack, or discontinuation, of services as a result of funding gaps, lack of confidential and safe spaces, mistrust in services as a result of previous negative experiences or mistreatment by service providers, and unavailability of on-site childcare discouraged survivors of violence from seeking help. Some also identified a lack of qualified practitioners as a primary barrier to seeking mental health services. Others suggested that legal or social barriers such as mental health stigma, fear of checkpoints for refugees, inability to leave the home as a result of the physical effects of violence, and spousal restrictions contribute to help-seeking behaviors. Additionally, while some participants noted a lack of knowledge about services, more were concerned with logistical and financial barriers such as transportation costs, distance to service providers, and inability to cover costs.

Nearly half of the women who did seek services in response to violence (47%) said the primary supportive factor was that the services were targeted to people of their background or status. This was a particularly important supporting factor among Syrian women (52%) but less so among Lebanese women (39%) (*p* < 0.001). Clear and accessible information was reported as the primary supporting factor for 15% of women, with significant differences between regions (20% Bekaa vs. 9% North) (*p* < 0.001). In FGDs, participants noted proximity, collocated services (“one stop shop”), and low- or no-cost as significant supportive factors, with participants in the Bekaa also mentioning provision of transportation as a supporting factor.

Neither the survey nor the FGDs found any potent relationship between individual coping mechanisms and improved mental health. The coping mechanisms that were reported as helpful were those that either changed a participant’s external environment (e.g., finding a job to increase income or secure housing), or that changed participants’ perception of, or psychosomatic response to, their environment (e.g., pain relievers or other medication). Other coping mechanisms such as crying about it or letting it out were described as “of no use” because the “sorrow remains inside”. One participant found “nothing beneficial”. Additionally, while very few participants reported prayer or religion as a helpful mechanism, one woman said, “I go up to the edge of the valley and I talk to God. I sit down, drink a cup of maté, smoke a cigarette, scream my lungs out, and then go back [home]”.

## 4. Discussion

Our study indicates that women who experience GBV were more likely to experience psychological distress, results that are consistent with other studies. Several researchers have noted that women who experience IPV are at an increased risk of experiencing severe mentally ill-health symptoms, including symptoms consistent with depression and PTSD [16,30,31]. While our study results are not based on diagnostic mental health scales, and thus do not indicate specific mental health disorders, they do indicate the need for more preventative GBV services and specialized mental health services for GBV survivors in order to minimize their contribution to distress. Additionally, both survey and qualitative data collection revealed that a significant number of this service population and sample also experienced environmental stressors that might exacerbate both GBV and mental health risks. A potent example of this was the significantly increased odds of experiencing severe distress among those who were propositioned for transactional sex, in comparison to those who did not (aOR 4.2; 95% CI 1.2–15.3; *p* ≤ 0.05). This form of sexual exploitation is often a last resort for those who are otherwise unable to secure basic needs and resources, and has been shown to be significantly associated with mentally ill-health in other studies [32]. Young girls who engaged in transactional sex reported feeling “unhappy, miserable, depressed, tied up, or tense” in one of these studies [32].

Forced child marriage was also associated with three times the likelihood of severe distress when controlling for all other family and relationship variables (aOR 3.0; 95%CI 1.9–20.8; *p* ≤ 0.05) but lost significance in the final model. This can be considered in the framework of the environmental stressors reported by study participants and the proven intersections between child marriage, mentally ill-health, and environmental factors such as economic security, immigration status, and feelings of safety [33,34].

Survey respondents represent the population whence the sample was derived: the Abaad psychosocial support service users. As noted above, the study sample excludes women who have engaged in Abaad’s case management services, thus rates of high-risk intimate and non-partner sexual violence are likely lower than they would be among the general population. Common features of the beneficiaries are environmental vulnerabilities, child marriage that is often entered without their consent, and early pregnancy that can further stress capacity and resiliency of households and communities. In addition to reportedly burdensome gender roles, economic precarity, marginalization due to refugee status, and social characteristics of refugee camps contribute to the prevalence and normalization of GBV, especially rape and IPV [35]. Stress related to separation from family members, uncertainty about the future, and lack of legal aid—common features among Syrian refugee populations in other settings [11]—compound psychological distress and anxiety and create a barrier for women to seek support services. Another reported barrier was a lack of awareness about GBV prevention and response services. Given our sample was from women who had engaged Abaad’s psychosocial support services and have increased access to the broader humanitarian network, it is likely that rates of help-seeking are higher in this sample than they would be among the general population, and especially among GBV survivors who have not accessed humanitarian assistance. A lack of census data and publicly available GBV data in Lebanon make comparisons difficult between them and broader populations, though greater lack of awareness among the general population may translate to an even larger need for demand creation and outreach by GBV and MHPSS service providers for women who have never accessed humanitarian services.

Rates of both GBV and mentally ill-health have increased since this data was collected in 2019, likely exacerbating the relationship between them and putting them at an increased risk of environmental stressors [36,37]. A global pandemic and stay-at-home orders, Lebanon‘s economic collapse, and the devastating Beirut Port Explosion have added significant pressures, increased vulnerabilities, and compromised mental health among both host and refugee communities [38,39,40], making the below recommendations increasingly relevant to Lebanon’s current climate.

There is a strong demand for consistent and culturally appropriate mental health care that engages the local community while incorporating psychosocial medical care [41,42], especially considering that social and familial networks are disrupted throughout conflict and displacement [11]. Programming should place an emphasis on access, taking into account gendered roles’ influence on daily lives of the women (e.g., need for childcare), and the need for clarity about who programs are for and associated costs, if any, as this was the primary supportive factor in service seeking listed among the survey group, and was a primary barrier among qualitative discussion respondents.

As far as we are aware, this was the first service-based research study on GBV and mental health in Lebanon, and thus will provide insight for other humanitarian actors on service prioritization, response to GBV survivors, and need for GBV and specialized mental health screening during health, psychosocial support, and other long-term services. While the study did not find any statistically significant association between individual coping mechanisms and mental health, future studies should explore the efficacy of individual coping mechanisms and service use on improved mental health among GBV survivors who live in potentially harmful environments, including that of conflict, poverty, or disasters.

The partnership between Abaad and the Global Women’s Institute (GWI) allowed for localized knowledge on GBV and the Lebanese context, and academic rigor in the design, implementation, and analysis of the study. Our collaboration centered capacity sharing in ‘decolonizing’ the research process, such that a member of the GBV service provision agency was supported to lead the research and its dissemination.

## 5. Conclusions

Service-based data proved valuable to better understand the gendered effects of war on populations’ physical and mental health and well-being, and to translate research to practice. We collaborated as a GBV service provider and a GBV-focused research institute to better understand how GBV and mental health issues intersect for women living as refugees, and those living in the increasingly vulnerable communities hosting them.

All experiences of GBV measured by the survey—intimate partner violence, non-partner violence, and forced/child marriage—were associated with severe psychological distress. Having ‘serious problems’ in any of five measured dimensions of environmental vulnerability was also associated with a significantly higher rate of severe psychological distress (compared to those who did not report a serious problem); and higher environmental vulnerability scores were significantly associated with increasing rates of psychological distress.

Barriers to seeking help included lack of awareness, not believing services were needed, and, for fewer women, fear of retaliation and husbands not allowing their wives to seek help. The primary supportive factor was that services targeted people of their background or status, particularly among Syrian refugee women, and clear and accessible information, proximity, collocated services, low- or no-cost, and provision of transportation.

These findings indicate several practical steps government, humanitarian, and health stakeholders can take to reduce the burden of GBV and mentally ill-health among women in Lebanon. Specifically, provide holistic, well-coordinated, low- to no-cost GBV and MHPSS services, with support for referral to tertiary care (i.e., in-patient or psychiatric care (keep in mind the care needed when prescribing psychiatric medication, as some FGD participants noted that psychiatrists are at times too quick to prescribe medications such as benzodiazepines)); conduct awareness sessions and campaigns on various forms of GBV, including victim-blaming and forced child marriage on community and service-provider levels; and engage local communities when developing mental health awareness and destigmatization campaigns.

It is necessary for service providers to hire and train qualified, sensitive, and accessible service providers for mental health and other non-GBV services (e.g., basic assistance and cash/food). These should target host, refugee, and migrant communities, ensure the availability of childcare services, and include contextualized economic empowerment activities.

Given the rapidly deteriorating political, economic, and public health situation in Lebanon since the data was collected, it is very likely that the both GBV and mental ill-health experiences have been exacerbated. Implementing these recommendations, with the participation of affected women, is vital to ensuring that a mental health crisis does not follow the COVID-19 pandemic, and that the ‘shadow pandemic’ of GBV is adequately addressed.

## Figures and Tables

**Table 1 ijerph-18-04500-t001:** Scales used to develop the quantitative tool.

Tool	Description	Validity in Lebanese Context
Humanitarian Emergency Settings Perceived Needs (HESPER) Scale [22]	Data Developed to rapidly assess perceived needs of populations in humanitarian settings in low-and middle-income countries.	HESPER has been field-tested in various humanitarian settings (including Jordan); pilot participants found items comprehensive and relevant, suggesting criterion and content validity [22].
COPE Inventory [23]	A list of coping mechanisms (e.g., “trusting in God” or using emotional support as means of coping).	Has not been validated in Lebanon or humanitarian settings
Kessler Psychological Distress Scale (K6) [24]	The K6 is used for screening or severity for mood or anxiety disorders. Responses to six items are scored on a five-point Likert scale with the range “none of the time” (1) to “all the time” (5), overall scores ranging from 6 to 30 [25]. Serious mental illness has a positive score between 19 and 30.	The Arabic K6 has been validated in Lebanon [26].
WHO “Situation of Women and Girls’ Health and Life Experience in Conflict Settings” survey questionnaire [27]	Adapted from the WHO Multi-Country Study on Women’s Health and Life Experiences to capture information on IPV and non-partner violence (NPV) in conflict-affected settings.	Adapted and piloted in humanitarian contexts by GWI, IRC and CARE in South Sudan [28].

**Table 2 ijerph-18-04500-t002:** Number and type of FGD participants.

	Number of People	Number of Groups
Service Providers	19	3
Community Leaders	32	4
Women Community Leaders	63	8
Total	114	15

**Table 3 ijerph-18-04500-t003:** Socioeconomic characteristics of women, overall and by nationality and region.

	All (%)	Lebanese (%)	Syrian † (%)	Beqaa (%)	North (%)
Age of respondents:					
Mean (Median)	38.3 (38)	41.3 (41.5)	36.5 (36)	36.3 (35)	41.0 (40)
18–24	17	13 *	19 *	23 *	8 *
25–34	25	17 *	29 *	26 *	23 *
35–44	27	28 *	26 *	23 *	31 *
45–54	19	24 *	16 *	15 *	25 *
55–65	13	18 *	10 *	13 *	13 *
Nationality:					
Lebanese	36	--	--	21 *	57 *
Syrian	64	--	--	79 *	43 *
Length of time living in Lebanon:					
5 years or less	20	--	20	23 *	11 *
more than 5 years	80	--	80	77 *	89 *
Educational attainment:					
less than primary	58	53 *	62 *	57	60
completed primary	27	26 *	28 *	28	26
secondary or higher	15	22 *	11 *	15	14
Participation in income generating activities:					
no	85	81 *	87 *	85	84
yes	15	19 *	13 *	15	16
Main source of household income:					
no income	5	5 *	6 *	4 *	6 *
money from own work	8	10 *	6 *	7 *	8 *
husband/partner	29	50 *	17 *	14 *	49 *
humanitarian aid	34	6 *	50 *	53 *	7 *
other source	25	30 *	21 *	21 *	29 *
N	969	352	617	562	407

* Chi-square statistic significant *p* ≤ 0.05. † Syrian statistics include (9) Palestinian respondents.

**Table 4 ijerph-18-04500-t004:** Marriage and and family characteristics of women, overall and by nationality and region.

	All (%)	Lebanese (%)	Syrian † (%)	Beqaa (%)	North (%)
Current partnership status:					
Married, living with partner	70	73 *	68 *	70 *	70 *
Partnered (married/engaged), living apart	6	5 *	7 *	6 *	6 *
Previously partnered	15	11 *	18 *	18 *	12 *
Single	9	12 *	7 *	6 *	13 *
Has ever partnered:					
no	9	11 *	7 *	6 *	13 *
yes	91	89 *	93 *	94 *	87 *
Has ever married:					
no	9	12 *	8 *	7 *	13 *
yes	91	88 *	92 *	93 *	87 *
Is currently partnered:					
no	24	22	25	24	24
yes	76	78	75	76	76
Age at first marriage:					
under 15	11	7 *	12 *	11	10
15–19	52	43 *	56 *	54	48
20–24	26	33 *	22 *	24	29
25 and over	12	16 *	10 *	10	14
Non-consensual marriage:					
no	85	86	84	86	84
yes	15	14	16	14	16
Marriage before age 18:					
no	61	70 *	56 *	57 *	66 *
yes	39	30 *	44 *	43 *	34 *
Forced child marriage:					
no	91	95 *	89 *	90	92
yes	9	5 *	11 *	10	8
Has ever been pregnant:					
no	14	19 *	11 *	17 *	9 *
yes	86	81 *	89 *	83 *	91 *
Age at first pregnancy:					
under 15	9	5 *	12 *	13 *	5 *
15–19	13	10 *	15 *	14 *	12 *
20–24	37	32 *	40 *	40 *	33 *
25 and over	41	53 *	34 *	34 *	50 *
Has children:					
no	2	2	2	2	1
yes	98	98	98	98	99
Number of children:					
1–2	25	19 *	27 *	28 *	20 *
3–5	51	57 *	48 *	47 *	55 *
>5	24	24 *	25 *	24 *	24 *
N	969	352	617	562	407

* Chi-square statistic significant *p* ≤ 0.05. † Syrian statistics include (9) Palestinian respondents.

**Table 5 ijerph-18-04500-t005:** Odds of experiencing physical and/or sexual IPV (lifetime) by significant factors.

	Crude Odds Ratio	95% CI	Adjusted Odd Ratio	95% CI	Adjusted Odd Ratio	95% CI
			Model I		Model II	
Works for money or engages in income generating activities						
No	1.0		1.0		1.0	
Yes	1.7 **	1.2–2.5	2.0 ***	1.3–3.1	1.9 **	1.1–2.6
Forced child marriage						
No	1.0		1.0		1.0	
Yes	2.7 ***	1.6–4.5	2.0 **	1.1–3.6	2.2 **	1.3–3.9
Partner’s Educational Attainment						
secondary or higher	1.0		1.0		1.0	
less than primary	1.7 **	1.0–2.7	1.7 **	1.0–2.7	1.6 **	1.0–2.4
completed primary	1.0	0.6–1.6	1.0	0.6–1.6	0.9	0.5–1.4
Has a serious problem with physical health due to illness, injury, or disability						
No	1.0		1.0		1.0	
Yes	1.8 ***	1.4–2.4	1.7 ***	1.3–2.2	1.4 **	1.0–1.9
Has a serious problem because she is separated from family members						
No	1.0		1.0		1.0	
Yes	1.7 ***	1.3–2.2	1.5 **	1.1–2.0	1.4 **	1.1–1.9
Has a serious problem due to lack of easy and safe access to a clean toilet						
No	1.0		1.0		1.0	
Yes	1.7 ***	1.3–2.3	1.5 **	1.1–2.0	1.5 **	1.1–2.0

*** *p* ≤ 0.001, ** *p* ≤ 0.05.

**Table 6 ijerph-18-04500-t006:** Odds of experiencing physical and/or sexual IPV (last 12 months), by significant factors.

	Crude Odds Ratio	95% CI	Adjusted Odd Ratio	95% CI	Adjusted Odd Ratio	95% CI
			Model I		Model II	
Age (10 year groups)						
55–65	1.0		1.0		1.0	
18–24	2.5 **	1.4–4.5	2.5 **	1.4–4.5	3.6 **	1.5–8.4
25–34	3.3 ***	2.0–5.6	3.3 ***	2.0–5.6	2.9 **	1.4–5.9
35–44	2.6 ***	1.6–4.3	2.6 ***	1.6–4.3	2.1 **	1.0–4.1
45–54	2.2 **	1.3–3.8	2.2 **	1.3–3.8	1.8	0.9–3.6
Partner is working						
Yes	1.0		1.0		1.0	
No	1.5 **	1.1–2.0	1.5 **	1.1–2.0	1.3 *	1.0–1.9
Forced child marriage						
No	1.0		1.0		1.0	
Yes	1.8 **	1.1–2.9	1.9 **	1.1–3.2	2.1 **	1.1–3.9
Number of children						
1 to 2	1.0		1.0		1.0	
3 to 5	1.4	0.9–2.1	1.6 **	1.0–2.4	0.9	0.5–1.6
more than 5	1.7 **	1.2–2.4	1.8 **	1.3–2.7	1.4	0.9–2.2
Has a serious problem with physical health due to illness, injury, or disability						
No	1.0		1.0		1.0	
Yes	1.5 **	1.1–2.0	1.4 **	1.1–1.9	2.0 ***	1.4–2.8
Has a serious problem because of lack of safety and security where she lives						
No	1.0		1.0		1.0	
Yes	1.7 ***	1.3–2.2	1.5 **	1.1–2.0	1.4 *	1.0–1.9

*** *p* ≤ 0.001, ** *p* ≤ 0.05, * *p* ≤ 0.10.

**Table 7 ijerph-18-04500-t007:** Odds of experiencing severe psychological distress in the past 30 days, by significant factors.

	Crude Odds Ratio	95% CI	Adjusted Odd Ratio	95% CI	Adjusted Odd Ratio	95% CI
			Model I		Model II	
Women’s age						
18–24	1.0		1.0		1.0	
25–34	4.1 ***	2.2–7.7	4.7 ***	2.4–9.3	3.8 **	1.5–9.2
35–44	5.8 ***	3.1–10.8	6.8 ***	3.5–13.3	3.7 **	1.6–8.9
45–54	2.5 **	1.3–4.6	3.6 ***	1.8–7.2	2.2	0.9–5.4
55–65	3.0 **	1.5–6.2	4.8 ***	2.1–10.6	3.2 *	1.1–9.0
Nationality						
Lebanese	1.0		1.0		1.0	
Syrian	3.3 ***	2.2–5.0	3.6 ***	2.2–2.7	2.5 **	1.4–4.5
Emotional violence, last 12 months						
No	1.0		1.0		1.0	
Yes	2.1 ***	1.4–3.3	2.1 ***	1.4–3.3	1.6	0.9–2.8
Forced child marriage						
No	1.0		1.0		1.0	
Yes	3.0 **	1.0–8.6	3.0 **	1.9–20.8	2.2	0.7–7.5
Offered money or ther favors in exchange for sex (non-partner)						
No	1.0		1.0		1.0	
Yes	6.3 **	1.9–20.7	6.3 **	1.9–20.8	4.2 **	1.2–15.3
Has a serious problem sue to safety and security where she lives						
No	1.0		1.0		1.0	
Yes	4.2 ***	2.7–6.6	2.8 ***	1.7–4.7	2.8 **	1.6–4.9
Has a serious problem due to physical health, illness, injury, or disability						
No	1.0		1.0		1.0	
Yes	2.9 ***	1.9–4.4	3.1 ***	1.9–5.1	2.4 **	1.3–4.4
Has any serious problems due to environmental vulnerabilities						
No	1.0		1.0		1.0	
Yes	11.7 ***	6.1–22.7	3.5 **	1.6–7.4	3.2 **	1.2–8.5

*** *p* ≤ 0.001, ** *p* ≤ 0.05, * *p* ≤ 0.10.

## Data Availability

Data is available in this DropBox file: https://www.dropbox.com/sh/dlc3od3zjeaye3x/AAAiUH7K_x2sZpxu4esSyonOa?dl=0 (accessed on 22 April 2021).

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
