# Peer review of "“I Go up to the Edge of the Valley, and I Talk to God”: Using Mixed Methods to Understand the Relationship between Gender-Based Violence and Mental Health among Lebanese and Syrian Refugee Women Engaged in Psychosocial Programming"

_ijerph, 2021, doi:10.3390/ijerph18094500_

Round 1
Reviewer 1 Report
Dear Author,
This interesting paper deserves to be published. The findings are based on the narratives of the participants and involve aspects that deserve to be shared: spirituality in relation to mental health.
I suggest improving the "Introduction" and including the theoretical concepts of the analysis variables.
I would like to continue reading articles like this. Success!
Author Response
Thank you very much for your helpful feedback! We integrated a brief theoretical framing in the introduction for the analysis variables. This theoretical framework is the social ecological model, as it helps us understand women's environmental factors as well as their experiences of mental health in a rapidly changing and vulnerable setting.
Reviewer 2 Report
I believe that the topic of this manuscript is an important content that tells a very sad and disastrous reality in Lebanon. I would like to suggest some revisions to this manuscirpt.
Specific comments:
1.Writing
The writing, structure and organization of the manuscript is in accordance with the guidelines.
2. Methods
There are no reliability values for the quantitative tools. Please add them.
3. Results
Please provide a table of the subjects’ general characteristics. Please change the headings of Tables 3 and 4 to 95% CI. The values in the tables and the values in the text may be different, so please check them.
Author Response
Thank you very much for your helpful feedback!
We added the reliability values (Cronbach's) for the quantitative tools that were used as scales, and provided an explanation for the ones that were used as single-item questionnaires (as opposed to scales).
We also added tables for women's characteristics, adjusted tables' headings to 95% CI, and double checked that the values in the tables and text were the same.
Reviewer 3 Report
Brilliant research was done, and a very clearly written paper. Please see some ideas for your considerations in the attachment.

Author Response
Thank you very much for your helpful feedback!
We made sure to remove Dr. Contreras' email from the abstract, added a footnote for the AOR model I and II to define the respective analysis, added a few sentences to the discussion about need for demand creation for services (i.e. lack of awareness as a barrier to services), and decreased the conclusion content by 20%.